# Advances in the Study of Common and Rare *CFTR* Complex Alleles Using Intestinal Organoids

**DOI:** 10.3390/jpm14020129

**Published:** 2024-01-23

**Authors:** Maria Krasnova, Anna Efremova, Diana Mokrousova, Tatiana Bukharova, Nataliya Kashirskaya, Sergey Kutsev, Elena Kondratyeva, Dmitry Goldshtein

**Affiliations:** 1Research Centre for Medical Genetics, Moscow 115522, Russia; krasnova.m@med-gen.ru (M.K.); mokrousova.d@med-gen.ru (D.M.); kashirskayanj@med-gen.ru (N.K.); kutsev@mail.ru (S.K.); kondratyeva_ei@med-gen.ru (E.K.); dvgoldshtein@med-gen.ru (D.G.); 2Moscow Regional Research and Clinical Institute (“MONIKI”), Schepkina Street, 61/2, 1, Moscow 129110, Russia

**Keywords:** complex allele, cystic fibrosis, *CFTR*, intestinal organoid, targeted therapy, CFTR modulators

## Abstract

Complex alleles (CAs) arise when two or more nucleotide variants are present on a single allele. CAs of the *CFTR* gene complicate the cystic fibrosis diagnosis process, classification of pathogenic variants, and determination of the clinical picture of the disease and increase the need for additional studies to determine their pathogenicity and modulatory effect in response to targeted therapy. For several different populations around the world, characteristic CAs of the *CFTR* gene have been discovered, although in general the prevalence and pathogenicity of CAs have not been sufficiently studied. This review presents examples of using intestinal organoid models for assessments of the two most common and two rare *CFTR* CAs in individuals with cystic fibrosis in Russia.

## 1. Introduction

Cystic fibrosis (CF) is caused by pathogenic nucleotide variants of the *CFTR* gene, which encodes the cystic fibrosis transmembrane conductance regulator (CFTR) protein. CF is a monogenic, autosomal recessive disease that causes complications in multiple organs due to insufficient levels of CFTR protein on the apical membrane of epithelial cells or its dysfunction. The *CFTR* gene encodes the ion channel that transports chloride ions across cell membranes. Mutations of the *CFTR* gene lead to dysregulation of chloride transport in the epithelial tissues of the lungs, pancreas, intestines, and other organs. A large number of factors, including the diversity of *CFTR* genotypes, modifier genes, and environmental factors, can cause significant variations in the phenotypic manifestations of cystic fibrosis in individuals [1].

The diagnosis of CF and selection of targeted therapy are complicated due to the wide variety of variants of the nucleotide sequence of the *CFTR* gene; more than 2000 variants are currently known [2] but only 719 of them have been proven to be pathogenic [3]. Most of the identified variants need to be further studied to determine their pathogenicity. *CFTR* sequencing continually identifies new variants for which the clinical significance must also be determined.

The diagnosis and administration of targeted therapies are also complicated by complex alleles of *CFTR*. CAs occur when there are two or more nucleotide variants on the same allele (cis-variants). Characteristic CAs have been found in several different populations of CF individuals around the world, although the general prevalence and pathogenicity of CAs have not been sufficiently studied. In the Russian Federation, the frequency rate of different cis-variants with F508del among CF individuals amounts to 6.6%, highlighting the importance of studying the presence of complex alleles to make a genetic diagnosis [4]. In Russian individuals, the complex alleles [L467F;F508del] and [S466X;R1070Q] are the most common [5]. The frequency rate of [L467F;F508del] amounts to 0.74%, while the rate is 0.46% for [S466X;R1070Q] from the total number of CF individuals in the Russian Federation according to the national register of 2021 [5]. At the same time, a study of the allelic frequency of [L467F;F508del] among F508del homozygotes showed that the rate is ~8% [6].

Similar to the most common Russian variant [L467F;F508del], a complex allele [F508del;I1027T] was identified in France; it has a frequency rate of >5% of chromosomes carrying the F508del mutation [7]. The I1027T variant is a polymorphism that does not cause CF. In a study by Baldassarri et al., the functional activity of the CFTR channel was assessed in individuals with [F508del;I1027T] [8]. The CFBE41o (human CF bronchial epithelial) cell line from a CF (F508del/F508del) patient was transfected with a plasmid carrying the CA [F508del;I1027T] [8]. The assessment of the CFTR protein’s functional activity showed that in the case of [F508del;I1027T], the CFTR channel function amounted to 23%, while in the control F508del it was 13%. According to the obtained data, it can be concluded that the I1027T variant can reduce the pathogenicity of the F508del variant [8].

In Italy, the CA [A238V;F508del] is relatively common, with an allelic frequency of 0.04 among individuals with the F508del variant (18/436). A study conducted on a group of individuals carrying the aforementioned CA that included 18 individuals showed that this CA has fewer general complications compared to control groups, although on the other hand it causes a more severe pulmonary phenotype and higher local and systemic inflammatory responses [9]. This suggests that the A238V variant enhances the pathogenicity of the F508del variant.

The invention of small molecules that restore the functional activity of the CFTR channel has revolutionized the pathogenetic treatment of CF. In the context of the development of CFTR modulators, it is necessary to carefully study the *CFTR* gene’s pathogenic variants in order to select an effective targeted therapy. The issue of the influence of CAs on the effectiveness of targeted therapies is poorly understood.

Currently, the FDA has approved four CFTR modulators—the potentiator ivacaftor (VX-770) and correctors lumacaftor (VX-809), tezacaftor (VX-661), and elexacaftor (VX-445). There are currently three types of correctors, which are classified depending on whether they suppress conformational defects at the boundaries between nucleotide binding domain 1 (NBD1) and transmembrane domains 1 and 2 (TMD1 and TMD2, type I), defects in the nucleotide binding domain 2 (NBD2, type II), or NBD1 defects caused by F508del (type III). For example, the components elexacaftor and tezacaftor included in the elexacaftor + tezacaftor + ivacaftor (ETI) triple drug are type III and type I, respectively. The effects of tezacaftor on restoring the functionality of CFTR are related to suppressing cotranslational misfolding, while tezacaftor binds within TMD1 in certain misfolded CF variants [10].

The binding sites for lumacaftor and tezacaftor are almost identical and include the amino acid residues TM1 (lumacaftor—K68, I70, N71, L73, R74, F77, F78, F81; tezacaftor—N71, L73, R74), TM2 (M152), TM3 (G194, L195, A198), and TM6 (T360, W361, S364, L365, and I368) [11,12]. Lumacaftor interacts with CFTR predominantly through van der Waals interactions, except for a salt bridge with residue K68. The polar half of lumacaftor extends outside the pocket, tethering the cytoplasmic ends of TM1 and TM6 together by interacting with residues 70–74 on TM1 and L365 and I368 on TM6. Instead of forming a salt bridge with K68, tezacaftor forms an H-bond with R74 [11].

The corrector elexacaftor (VX-445), which targets TMD2 (TMH10-11) and the N-terminal Lasso segment, restored the WT-like folding efficiency (~36%) of F508del-CFTR [12]. Elexacaftor interacts with the CFTR protein through both hydrogen bonding and by forming a salt bridge with the amino acid residue R1102 located in TM11 [13]. In TM11, elexacaftor also interacts with the W1098 residue. In the Lasso motif, elexacaftor forms hydrogen bonds with residues S18 and R21 [12]. It should be noted that all correctors show their effectiveness in the presence of the potentiator ivacaftor (VX-770), which binds to TM4, TM5, and TM8, enhancing the opening of the CFTR channel [13].

For example, the most common variant F508del in trans-position leads to disruption of the folding of the CFTR protein and prevents its transport to the plasmatic membrane. The F508del variant is curable with triple-targeted ETI therapy, as well as a double-targeted therapy with ivacaftor + lumacaftor [14].

Each pathogenic variant in the CA composition can affect various stages of CFTR protein biogenesis; for example, the protein may not be translated or misfolded, meaning its conductivity is impaired. Thus, each *CFTR* nucleotide variant has a different effect on the translated protein. Known pathogenic variants of the *CFTR* gene that are sensitive to CFTR modulators (included in the instructions for targeted drugs) as part of the CAs may lead to the absence of a pharmacological response to the targeted therapy.

For example, CFTR correctors are ineffective if the action of the cis-variants leads to a decrease in CFTR protein expression or shielding of the binding site of the corrector by changing the protein conformation.

Thus, CAs complicate the classification of *CFTR* variants and require additional studies to determine their pathogenicity and modulatory effect on the targeted treatment. This review summarizes the results for all CAs we studied using intestinal organoids, namely [L467F;F508del], [S466X;R1070Q], [E217G;G509D], and [F1052V;1367del5]. A schematic representation of the localization of the CFTR variants covered in the review is shown in Figure 1.

## 2. Intestinal Organoids as a Model for Studying CFTR Channel Function

Organoids are self-organizing, three-dimensional, organ-specific clusters derived from stem cells that are capable of maintaining the cellular heterogeneity and functional properties of the organ from which they are derived (e.g., brain, retina, liver, or intestine). Intestinal organoids (IOs) comprise of the most widely studied organoid types to date. For the first time, IOs were derived from a intestinal stem cell expressing Lgr5 [15]. Intestinal organoids are capable of recapitulating the structure, cellular composition, and functional features of the intestinal epithelium [12]. Domains similar to the crypts and villi of the intestinal epithelium are distinguished in the IOs. As in the intestinal epithelium, the “crypt” of intestinal organoids also contains stem cells and Paneth cells. The “villus” of IOs consists of differentiated cells (intestinal enterocyte, goblet, and enteroendocrine cells), which also correlates with the structure of the villus of the intestinal epithelium [15,16]. It was experimentally shown that all types of cells in IOs perform the same functions as their counterparts in the intestinal epithelium; brush-bordered enterocytes absorb nutrients, goblet cells produce mucus, Paneth cells secrete antimicrobial peptides and factors for maintaining the stem cell niche, and enteroendocrine cells secrete hormones. Due to such similarities with the intestinal tissue, organoids are also called mini-guts [16].

The emergence of intestinal organoid cultures has made it possible to model the intestinal epithelium in vitro and overcome a number of limitations inherent to transformed cell lines, such as their inability to reproduce the cellular diversity of the intestinal mucosa in a cell line [17]. The cellular composition of intestinal organoids also depends on the presence of a concentration gradient of growth factors in the culture medium. Cultures of organoids of the human large intestine (colon), in contrast to organoids of the small intestine, due to the absence of a concentration gradient of Wnt-3A growth factor in the culture medium under standard culture conditions, contain predominantly LGR5+ stem cells [18].

The surge of various studies on IO cultures over the past decades has revolutionized the field of biomedicine. Intestinal organoid technology is used in a wide range of research fields, including disease modeling [17], the development and testing of drugs and vaccines [19], studies of the interaction of the intestinal microbiota with the host organism [20,21], biomolecule delivery [22,23], and intestinal developmental biology [24]. Research using genome editing technologies is carried out on IOs, since organoids represent a convenient model for genetic manipulations to model hereditary pathologies and study DNA repair and the functional roles of individual genes through targeted mutagenesis [17,25,26].

Organoids can be obtained from two sources of stem cells, namely from tissue biopsies containing adult stem cells (ASC) and from pluripotent stem cells (iPSCs (induced pluripotent stem cells) or ESCs (embryonic stem cells)) [27].

Such optimized methods make it possible to obtain three-dimensional structures of organoids that reflect the microarchitecture of villi and crypts of the intestine and are capable of self-renewal and self-organization over long periods of time [15,18,28]. The organoid culture involves Matrigel (a commercial reagent enriched with components of the extracellular matrix, ECM) and a culture medium supplemented with a factor cocktail to promote the differentiation of stem cells into intestinal cells [18,29,30]. The extracellular matrix provides the environment with biochemical signals necessary for the formation of 3-D structures, their attachment, and the growth and differentiation of stem cells. The long-term maintenance of organoid culture is highly dependent on the components of the culture medium, which must trigger stem cell niche signaling pathways to maintain stem cell functions and promote their proliferation or differentiation into tissue-specific cell types [16]. The primary growth factors for culturing IOs are Wnt-3a (W), epidermal growth factor (EGF) (E), Noggin (N), and R-spondin-1 (R)—collectively called WENR [18,29,30].

To select a targeted therapy for CF individuals, the functional activity of the CFTR channel is assessed, which is carried out in vitro on individual’s intestine-derived organoids obtained from rectal biopsies. This method relies on a forskolin-induced swelling (FIS) assay, which allows the individual testing of targeted drugs with high throughput rates [31,32,33,34,35]. The principle of this assay consists of adenylate cyclase activation by forskolin in epithelial cells, which induces CFTR channel opening and ion secretion into the lumens of organoids. Organoid swelling induced by forskolin is selectively associated with CFTR channel function, which has been proven using CFTRinh-172 or GlyH-101 and *Cftr* gene knockout [28]. The forskolin-induced swelling (FIS) of patient-derived intestinal organoids is an in vitro biomarker that quantifies CFTR-dependent ion transport into the organoid’s lumen [28,31,32] and may provide a more accurate assessment of CFTR function compared with other methods. Some studies have shown that FIS results correlate with the sweat chloride concentration (SCC), intestinal current measurements (ICMs), and clinical aspects of the disease [31].

## 3. Results of Intestinal-Organoid-Based Study of the Complex Alleles [L467F;F508del] and [S466X;R1070Q] Common in Russian Individuals

Kondratyeva et. al. described a case of an individual who was initially diagnosed with the F508del/F508del genotype and correspondingly treated with the tezacaftor + ivacaftor combination, which is prescribed for the treatment of cystic fibrosis in patients over 6 years of age, including those with the F508del/F508del genotype. Due to the absence of changes in the individual’s condition when using targeted therapy, as well as the absence of changes in SCC and ex vivo CFTR channel function in rectal biopsies when measuring the short-circuit current density using the ICM method, the authors assumed the presence of the CA. Further studies of the forskolin-induced swelling of patient-derived IOs revealed a weak response to tezacaftor + ivacaftor (VX-661 + VX-770) application (Figure 2). The full sequencing of all *CFTR* exons identified the second pathogenic variant in exon 11, which causes L467F amino acid substitution in the cis position [36].

An analysis of the allelic frequency of [L467F;F508del] in the Russian Federation showed that it amounts to ~8% among F508del homozygotes [6].

The clinical manifestations in CF individuals bearing the [L467F;F508del]/F508del genotype compared with the F508del/F508del group, according to the literature, did not differ significantly in terms of the pulmonary function indicators, nutritional status, and microbiological profile [6]. As a particular feature, the authors have noted characteristic liver damage in patients with this CA genotype compared to the control group.

Cultures of IOs obtained from individuals bearing the [L467F;F508del] complex allele with the F508del variant in a compound heterozygous position and a healthy donor (wt/wt CFTR) are shown on Figure 3. The CF-derived cultures manifest morphological features of a dysfunctional CFTR channel, including characteristic thickening of the walls of the organoids, the absence of the lumen, and an irregular shape, in contrast to the culture derived from a healthy individual. The organoids with wt/wt CFTR possess a regular spherical shape, thin walls, and a clearly defined lumen (Figure 3).

The combinations of VX-770 + VX-809 and VX-770 + VX-661 effectively increased the amount of functional CFTR on the membranes of epithelial cells in the control F508del/F508del culture (Table 1), as the organoids showed 61.6% and 45.8% increases in their size, respectively. At the same time, in the case of the [L467F;F508del]/F508del genotype, the effect of these combinations of drugs was two-fold lower (33.6% for VX-770 + VX-809 and 24.7% for VX-770 + VX-661). According to these results, the therapeutic effect of this targeted drug treatment is expected to be insignificant or completely absent. On the other hand, upon simultaneous exposure to the three-component drug VX-770 + VX-661 + VX-445, the [L467F;F508del]/F508del organoids responded by swelling and showed a >100% size increase that indicated the effective recovery of the CFTR channel’s functional activity (Table 1, Figure 2). The obtained responses were comparable to the control ones.

Thus, using intestinal organoids, it was found that [L467F;F508del]/F508del individuals would be susceptive to triple ETI therapy in clinical practice [6,36], the effect that is associated with the presence of one F508del allelic variant in their genotype. For homozygous [L467F;F508del]/[L467F;F508del] individuals, the triple ETI combination is expected to be ineffective, as the [L467F;F508del] CA leads to a complete loss of functional CFTR protein [6,36].

To assess the pathogenicity of the [L467F;F508del] variant, Sondo et al. performed an in vitro study of the expression of L467F and F508del variants in FRT and CFBE41o-cells [37]. In F508del-CFTR-transfected cells, the immature core-glycosylated CFTR protein was predominant. After applying elexacaftor + tezacaftor correctors together, the level of fully glycosylated mature CFTR protein was increased. Cells transfected with L467F-CFTR expressed both immature and mature forms of the CFTR protein. The treatment with elexacaftor + tezacaftor increased the expression of the mature form. Consistent with the functional data, cells expressing [L467F;F508del]-CFTR predominantly expressed the immature form of the protein. The treatment with elexacaftor + tezacaftor led to a weak increase in the mature CFTR protein form in CFBE41o cells, while no positive effect of the correctors was observed in FRT cells [37].

**Table 1 jpm-14-00129-t001:** Alterations of organoid size with CAs after 60 min treatment of 5 μM forskolin and CFTR modulators (each at a concentration of 3.5 μM) (results of our previous studies [6,38,39]). The initial organoid area before forskolin stimulation (0 min) was taken as 100%; data are presented as means ± SD (%); F508del/F508del—control. Calculations were based on the results of three independent experiments obtained from a single culture.

	Genotype	Fsk	Fsk + VX-770	Fsk + VX-809	Fsk + VX-770 + VX-809	Fsk + VX-770 + VX-661	Fsk + VX-770 + VX-661 + VX-445
1	[L467F;F508del]/ F508del	107.9 ± 3.6	114.3 ± 3.2	114.7 ± 4.5	133.6 ± 12.1	124.7 ± 9.6	214.3 ± 22.6
2	[S466X;R1070Q]/ CFTRdele2,3	102.0 ± 0.7	102.8 ± 1.9	103.2 ± 2.3	104.6 ± 4.2	104.2 ± 3.4	105.1 ± 4.6
3	[E217G;G509D]/ F508del	151.2 ± 4.0	233.5 ± 16.6	214.8 ± 13.0	231.2 ± 18.6	216.6 ± 19.4	229.2 ± 12.5
4	[F1052V;1367del5]/ c.1209 + 2T > C	101.0 ± 0.4	103.0 ± 1.0	103.8 ± 0.8	102.9 ± 0.2	105.7 ± 0.1	111.7 ± 0.9
5	F508del/F508del	104.6 ± 4.2	111.7 ± 8.3	118.0 ± 9.2	161.6 ± 6.9	145.8 ± 9.7	220.9 ± 8.2

The CA [S466X;R1070Q] was previously studied in an individual with the [S466X;R1070Q]/CFTRdele2,3 genotype [38]. [S466X;R1070Q] was show to be pathogenic, and according to the RF Register 2021, occurs in Russia at a frequency rate of 0.46% among CF individuals, ranking 17th among 233 *CFTR* pathogenic variants described in the Russian Federation [5]. The S466X variant independently causes the premature termination of translation at position 466, with a frequency rate of 0.18% among CF individuals, ranking 27th according to the RF Register 2021. The R1070Q variant is classified as “mild” because the individuals bearing R1070Q are characterized by preserved pancreatic function. This variant causes the replacement of arginine by glutamine at position 1070; in Russia it occurs at a frequency rate of 0.08% and is in 54th place [5]. Notably, in the Russian Federation, each of these variants individually occurs with much lower frequency than as part of the [S466X;R1070Q] CA.

In our study [38], the individual with [S466X;R1070Q] bore the second allele CFTRdele2,3, which causes the deletion of exons 2 and 3 and is considered ‘severe’, ranking second among Russian *CFTR* variants after F508del [5]. The disease was characterized by the progressive degradation of lung function. The complete loss of the functional activity of the CFTR channel was evidenced by the lack of response to forskolin (observed using both ICM and FIS assays). In addition, it turned out that [S466X;R1070Q] is not sensitive to any of the tested targeted drugs and causes a complete loss of functional CFTR protein (Table 1, Figure 4) [38].

## 4. Assessment of Rare *CFTR* Complex Alleles Using Intestinal Organoids

For the first time, the [E217G;G509D] CA was described in a Russian individual with CF at the Federal State Budgetary Scientific Institution Research Center for Medical Genetics. It had not been previously described in any of the international databases, and the pathogenicity of this variant was investigated in [39]. The [E217G;G509D] CA includes a polymorphic E217G variant (c.650A > G, p.Glu217Gly) and G509D variant (c.1526G > A, p.Gly509Asp). According to previous data, the missense E217G variant (a replacement of glutamic acid with glycine) is considered a genetic variant of uncertain significance. The G509D variant is rare and results in a glycine to aspartic acid substitution in the nucleotide-binding domain NBD1 (Figure 1). In 2023, Kondratyeva et al. [39] carried out a comprehensive study of the rare [E217G;G509D] allele and CF’s pathogenesis and conducted an assessment of the clinical outcomes, with potential implications for genetic counseling and the personalized prescription of CFTR modulators. In [E217G;G509D]/F508del individuals, complications are primarily observed in the lungs, manifested by chronic bronchitis and the appearance of pathogenic flora, while the pancreas sufficiency is retained. IO-based studies have shown the high residual functional channel activity in this rare CA (Table 1, Figure 4); upon forskolin treatment in the absence of CFTR modulators, swelling of [E217G;G509D]/F508del organoids is observed due to CFTR activation. Since the second F508del allele causes almost a complete loss of functional CFTR [6,39], the organoid response can only be attributed to the [E217G;G509D] allele. All four targeted drugs approved by the FDA for CF were tested, and each had a positive effect of increasing the functional CFTR protein level (Table 1). Obtained in IOs, the results for [E217G;G509D] correlate with the ICM and SCC data and show that this CA belongs to the “mild” class and is susceptible to targeted therapy [39].

In 2022, the [F1052V;1367del5] CA was studied in an individual bearing the [F1052V;1367del5]/c.1209 + 2T > C genotype using an IO model. The complex allele [F1052V;1367del5] had not been previously described. The F1052V variant independently leads to the replacement of phenylalanine with valine at position 1052; in the Russian Federation, this variant is recorded at a frequency rate of 0.01% [5], with uncertain clinical significance. A total of 33 individuals with this variant are registered in the CFTR2 database [3]. The pathogenic variant 1367del5 (p.Asp415X) leads to the appearance of a stop codon that terminates translation at position 415 of the protein; it is found in Russia at a frequency rate of 0.42% [5]; 99 individuals with this variant are registered in the CFTR2 database [3]. In combination with a pathogenic mutation on the second parental allele, this variant leads to severe cystic fibrosis with pancreatic insufficiency [3,5]. It was expected that the F1052V variant, listed in the specifications of the targeted drugs Kalydeco^®^, Symdeco^®^, and Trikafta^®^, would cause a positive response to stimulation with forskolin upon exposure to CFTR modulators [40]. However, our studies have shown that the [F1052V;1367del5]/c.1209 + 2T > C genotype is associated with the complete disruption of the CFTR function, and targeted drugs do not restore CFTR’s functional activity (Table 1, Figure 4). A minor response (11% increase in organoid size) was observed only with simultaneous exposure to VX-770 + VX-661 + VX-445 and 5 μM of forskolin, although this effect cannot be the basis for recommending a targeted therapy to the individual.

## 5. Conclusions

The complex alleles in the *CFTR* gene contain at least two variants in the cis-position, and each variant can affect different steps of the CFTR protein biogenesis process, altering the overall pathogenicity of the allele [41,42]. Often, each variant in a complex allele can itself be a polymorphism and may not cause CF, although their joint pathogenic effect can affect the clinical manifestations of the disease [41,42,43,44,45]. The effects of complex alleles on the phenotype also depend on the second parental allele variant (in trans position). Individuals with CF should be tested for complex alleles of the *CFTR* gene, especially if individuals with the same genotypes have differences in disease manifestations or in response to the targeted therapy. Complex alleles may influence the treatment efficacy by reducing the impact of the CFTR modulator and increasing the diversity of pathogenic variants that must be considered when selecting the optimal therapy.

Thus, it was previously shown that additional cis-variants can influence the severity of clinical manifestations and the responses to therapy with CFTR modulators in individuals with CF [6,36,38]. In this work, we highlight the importance of diagnosing additional variants in the cis-position by demonstrating the results of our own studies conducted on IOs of individuals with complex *CFTR* alleles, occurring at different frequency rates in the Russian Federation. We assessed the residual functional activity of the CFTR channel and the effectiveness of targeted therapies. Even with CF individuals bearing an uncharacterized genotype, the FIS assay using IOs makes it possible to study the residual functional activity of the CFTR channel, as well as to determine the individual effects of targeted drugs on restoring the CFTR function.

If we evaluate the functional activity of the CFTR channel, we can conclude that the F508del variant is “milder” compared to CA [L467F;F508del], [S466X;R1070Q], and [F1052V;1367del5], which can be classified as class I-II and are more severe compared to [E217G;G509D] (Table 2).

The results of the FIS assay provide the basis for prescribing targeted therapies to CF individuals. As exemplified in the Russian Federation by the common [L467F;F508del] CA, it was shown that additional cis-variants affecting the folding of the CFTR protein can cause resistance to therapy with lumacaftor and tezacaftor correctors [6,36]. It was shown that the triple-targeted drug ETI is effective in the presence of the [L467F;F508del] CA in a compound heterozygous condition with F508del, indicating that the protein produced from the allele carrying the F508del variant is sufficient for an effective targeted therapy. It is expected that intestinal organoids with the [L467F;F508del]/[L467F;F508del] genotype will not respond to forskolin and targeted drugs. In a study by Baatallah et al. on the HEK293 cell line transfected with plasmids with a normal (WT) genetic variant of the *CFTR* gene, as well as the mutant variants L467F and F508del, Western blotting showed that the amounts of functional protein (mature, fully glycosylated) for the indicated variants are 86% (WT), 40% (L467F), and 22% (F508del). Thus, in the presence of the F508del/F508del genotype, both correctors and potentiators have a target for action. However, in the complex allele [L467F;F508del], an additional mutation L467F (which reduces the amount of mature protein by half compared to WT) almost completely suppresses the formation of functional CFTR protein. In this case, CFTR modulators do not have a target for action and the targeted therapy cannot be effective [46]. In Sondo’s study performed on transduced CFBE41o and FRT cells, [L467F;F508del] is classified as “severe” [37].

In the case of [S466X;R1070Q], although the R1070Q variant is considered as “mild” [5,42], S466X can produces the truncated CFTR variant that does not have the elexacaftor TM11 binding site [13] or only mRNA that undergoes degradation through the nonsense-mediated decay (NMD) mechanism. Additionally, [S466X;R1070Q] causes the complete loss of CFTR channel functional activity. In addition, it was found that this CA does not respond to any of the tested targeted drugs [38].

Furthermore, [F1052V;1367del5] is not treatable with ETI because the 1367del5 (c.1243_1247del) variant results in a 5-nucleotide deletion and is a class I mutation. In this case, there is likely to be a complete disruption of CFTR protein synthesis. In the absence of the protein, therapies with any currently known targeted drugs will be ineffective. The F1052V cis-variant as part of this CA does not contribute to the functional activity of the CFTR protein. The trans-variant c.1209 + 2T > C located in intron 9 of the *CFTR* gene is poorly studied and rare, and it has been described as a site-splicing variant [47]. The lack of response to ETI therapy allows us to assume that the c.1209 + 2T > C variant belongs to group of “severe” *CFTR* variants, although additional functional studies are required.

The [E217G;G509D] CA is the “mildest” of four CAs studied and exhibits sensitivity to all CFTR modulators.

In summary, *CFTR* gene CAs are relatively common features that need to be taken into account and cause the need to search for additional variants in the cis-position in case of a non-canonical individual response to the recommended therapy. For the personalized selection of CFTR modulators, especially in the case of rare CAs, it is necessary to conduct an FIS assay using the patient-derived IO culture.

Since the frequency of *CFTR* gene CAs is very high, it is necessary to sequence the *CFTR* gene in all newborns at the time of diagnosis in the future and to clarify the genetic diagnosis for those individuals who were diagnosed with CF previously.

New data will contribute to the evaluation of polymorphisms in the CAs to the phenotypic manifestations of CF, as well as to identifying new pathogenic CAs and changing the classification and revising the pathogenicity of previously described variants and their clinical manifestations. The obtained data will allow us to evaluate the effectiveness of targeted therapies and the correlation of the genotype with the development of various complications (impaired lung function, frequency of infection of the respiratory tract by microbial pathogens, pancreatic insufficiency). Obtaining a detailed description of each variant of CA will improve the genetic counseling for further development of the treatment strategies for corresponding individuals.

## Figures and Tables

**Figure 1 jpm-14-00129-f001:**
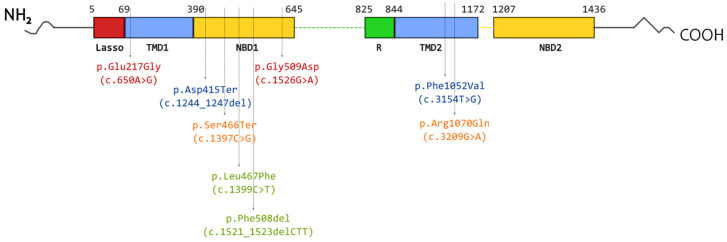
Schematic representation of the locations of variants included in the studied complex alleles in the CFTR protein. Font color indicates variants found on one allele: green—[L467F;F508del]; orange—[S466X;R1070Q]; red—[E217G;G509D]; blue—[F1052V;1367del5].

**Figure 2 jpm-14-00129-f002:**
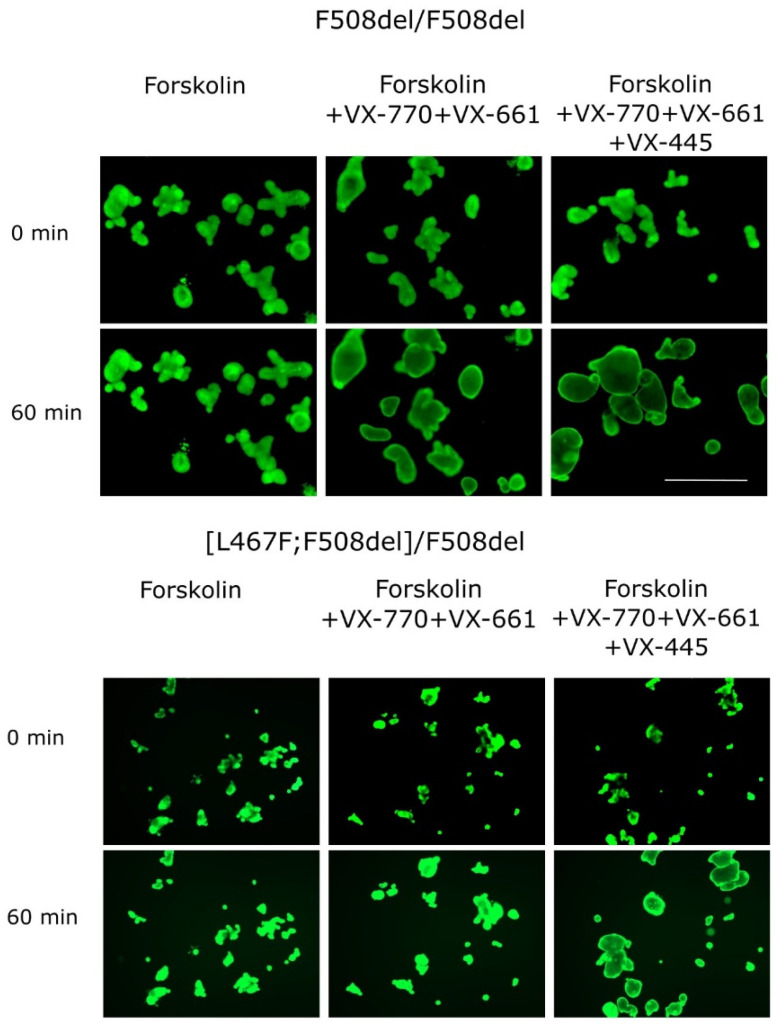
Images of intestinal organoids obtained from [L467F;F508del] CA individuals and from the F508del/F508del control culture before forskolin (5 μM) and targeted drug application (0 min) and 60 min after the treatment. Calcein staining (0.84 µM, 1 h), 5X objective, scale bar = 500 µm.

**Figure 3 jpm-14-00129-f003:**
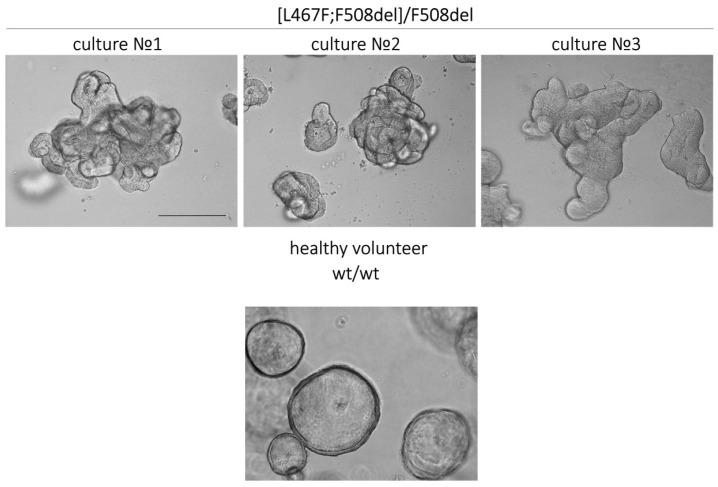
Morphological features of intestinal organoid cultures derived from three CF individuals with the [L467F;F508del]/F508del genotype and a healthy volunteer. Scale bar: 200 µm.

**Figure 4 jpm-14-00129-f004:**
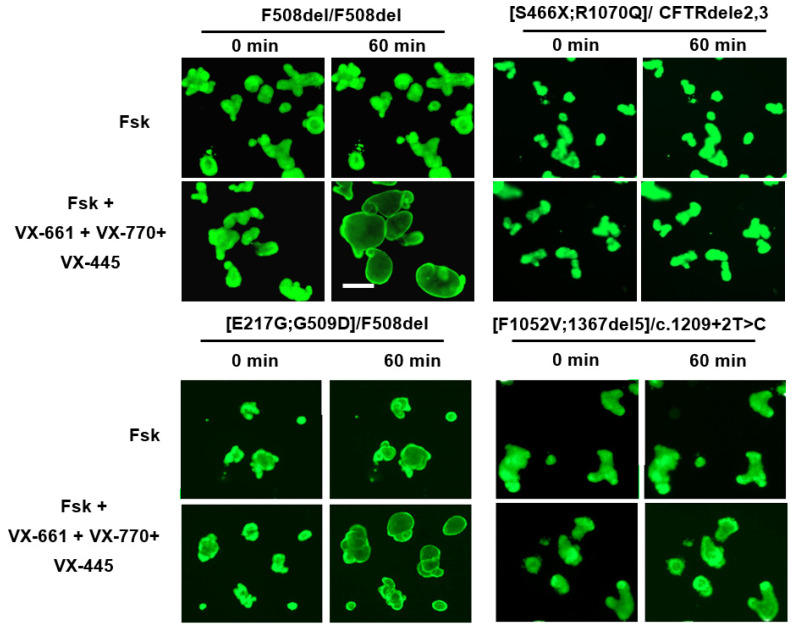
Study of residual CFTR channel function (forskolin response) and therapeutic effect of the triple ETI (VX-770 + VX-661 + VX-445, each at a concentration of 3.5 μM) combination in IOs derived from individuals with different CAs. F508del/F508del—control; forskolin concentration—5 µM; scale bar—200 µm.

**Table 2 jpm-14-00129-t002:** Summary data describing the effects of the studied CAs on CFTR protein expression.

Variant	Class	Reference
[L467F;F508del]	I or II	[6,36]
[S466X;R1070Q]	I	[38,42]
[E217G;G509D]	IV–VI	[37]
[F1052V;1367del5]	I	Results of own research, data not published

## Data Availability

The data from the current article are available from the corresponding author upon reasonable request.

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
