# Peer review of "Advances in the Study of Common and Rare CFTR Complex Alleles Using Intestinal Organoids"

_jpm, 2024, doi:10.3390/jpm14020129_

Round 1

Reviewer 1 Report

Comments and Suggestions for Authors

This short review summarizes recent studies on the effects of CA and CFTR modulators on them. Although I have concerns about potential similarities between the results presented in Figure 1 and Figure 2 and those from reference 33, they broadly discuss recent findings regarding CA of CFTR.

1. The binding site of Trikafta (VX-661, VX-445, VX-770) has recently been revealed. It would be nice to have a little more discussion about the difference in the Trikafta susceptibility due to the CA from the perspective of Trikafta's mechanism of action. For instance, S466X produces the truncated CFTR variant which has no binding sites of Trikafta. It would be nice to explain more why Trikafta is not effective on the CA [F1052V;1367del5]/c.1209+2T>C.

2. It would be nice to explain how c.1209+2T>C (intron variant) affects CFTR expression.

3. It would be nice to add a table summarizing how each CA affects CFTR expression in combination with ∆F508 mutation (e.g., class I defects, etc) if possible.

Author Response

This short review summarizes recent studies on the effects of CA and CFTR modulators on them. Although I have concerns about potential similarities between the results presented in Figure 1 and Figure 2 and those from reference 33, they broadly discuss recent findings regarding CA of CFTR.

Response: Dear reviewer, we would like to bring to your attention that Figures 2, 3 (new numbering of figures) and 5 from reference 33 (now it is reference 36) show various organoid images from different patients with CA [L467F;F508del]. It is important to note that despite the differences in CF individuals, the responses to forskolin and modulators are expectedly similar. This is due to the fact that both this article and reference 36 feature patients with the same CFTR genotype.

  1. The binding site of Trikafta (VX-661, VX-445, VX-770) has recently been revealed. It would be nice to have a little more discussion about the difference in the Trikafta susceptibility due to the CA from the perspective of Trikafta's mechanism of action. For instance, S466X produces the truncated CFTR variant which has no binding sites of Trikafta. It would be nice to explain more why Trikafta is not effective on the CA [F1052V;1367del5]/c.1209+2T>C.

Response 1: We greatly appreciate for your comments! We have added some information about binding sites and mechanisms of action of CFTR modulators, please, see the lines 71-97 and 342-381.

  1. It would be nice to explain how c.1209+2T>C (intron variant) affects CFTR expression.

Response 2: Dear Reviewer, thank you for your comment! We provide a description of this variant  in the Conclusion section, lines 378-381.

  1. It would be nice to add a table summarizing how each CA affects CFTR expression in combination with ∆F508 mutation (e.g., class I defects, etc) if possible.

Response 3: We are greatly appreciating for your suggestion; however, we have added a summary table of the variants we studied before and describe how CA affects CFTR expression according to the generally accepted classification of CFTR mutations (Table 2, lines 365). We did not include variants with F508del, but we additionally wrote on the lines 98-101 that the presence of the F508del variant in trans-position leads to disruption of the folding of the CFTR protein and prevents its transport to the plasmatic membrane.

  1. Figures are completely missing. Please add schematic drawings of where the different alleles are situated and what the consequences are.

Response 4: We appreciate your comments very much and have added a schematic drawing (Figure 1), showing the location of the studied complex alleles in the domain organization of the CFTR protein at line 117-119.

Reviewer 2 Report

Comments and Suggestions for Authors

Abstract

Page 1, Line 11 “rise the need for ….” grammatically incorrect. Please rephrase.

Introduction

Page 1, line 16 “patients with cystic fibrosis”. CF community in US prefers to call individuals with cystic fibrosis rather patients. As you would understand term patients label them as sick all the time and may be even hospitalized. But you are referring mostly to the variant they harbor rather than talking about their clinical course in this review. This is just a suggestion. If you agree to change please make correction at all places it is mentioned in the review.

Page 1, Line 26. Please check CFTR does not transport water ions through its channel. These are different channels like aquaporins that transport water in response to movement of chloride through CFTR and sodium ions through ENaC.

Page 2, line 49. Change exclamation mark (!) to period (.)

Page 2, line 60. After the word “variant” there is a weird typo erro before the reference. Please remove.

Page 2, line 56. Reference 6 is not correct for the findings described. You are talking about complex allele in CFTR with reference to F508del and I1027T, but the reference talks about COVID.

Furthermore, I do not agree with your conclusion on this statement that I1027T reduces the pathogenicity of F508del. Check revertants for this kind of conclusion like 4R and G550E published in literature from amaral and shepperad labs.

After this I am hesistant to review the article further. Please make sure every reference is correct and statements are supported by correct literature.

Comments on the Quality of English Language

requires some editing

Author Response

  1. Abstract

Page 1, Line 11 “rise the need for ….” grammatically incorrect. Please rephrase.

Response 1: We greatly appreciate for your comments! We have rephrased.

  1. Introduction

Page 1, line 16 “patients with cystic fibrosis”. CF community in US prefers to call individuals with cystic fibrosis rather patients. As you would understand term patients label them as sick all the time and may be even hospitalized. But you are referring mostly to the variant they harbor rather than talking about their clinical course in this review. This is just a suggestion. If you agree to change, please make correction at all places it is mentioned in the review.

Response 2: We greatly appreciate your suggestion and agree that in this case, it is more appropriate to use the term "individuals" instead of "patients". We strive to use precise and accurate language in all of our communications, and your input helps us to achieve this goal. Thank you again for bringing this to our attention.

  1. Page 1, Line 26. Please check CFTR does not transport water ions through its channel. These are different channels like aquaporins that transport water in response to movement of chloride through CFTR and sodium ions through ENaC.

Response 3: We greatly appreciate for your comments! We have checked and changed the sentence.

  1. Page 2, line 49. Change exclamation mark (!) to period (.)

Response 4: We have changed.

  1. Page 2, line 60. After the word “variant” there is a weird typo error before the reference. Please remove.

Response 5: We have removed

  1. Page 2, line 56. Reference 6 is not correct for the findings described. You are talking about complex allele in CFTR with reference to F508del and I1027T, but the reference talks about COVID.

Response 6: Thank you for taking the time to carefully proofread our manuscript. We would like to acknowledge that we have not provided the full title of the cited article; we made a mistake. The article (now Reference 8) is titled “CFTR gain and loss of function alleles are associated with clinical outcomes of COVID-19.” We would like to clarify that the described CA [F508del;I1027T] was actually studied in the link that we provided. We hope that this information will help to clarify any confusion that may have arisen as a result of the name of the reference.

Once again, we appreciate your efforts in reviewing our manuscript and thank you for bringing this issue to our attention.

  1. Furthermore, I do not agree with your conclusion on this statement that I1027T reduces the pathogenicity of F508del. Check revertants for this kind of conclusion like 4R and G550E published in literature from amaral and shepperad labs.

Response 7: We would like to address your concern regarding the reference in question. The conclusions we have done were based on the article by Baldassarri, M., Zguro, K., et al. (2022) titled "Gain- and Loss-of-Function CFTR Alleles Are Associated with COVID-19 Clinical Outcomes," published in Cells, 11, 4096 (Reference 8). The article can be accessed at https://doi.org/10.3390/cells11244096. Regarding Figure 3 at this article, the authors describe the results obtained using the expression of CFTR protein in a heterologous expression system. Specifically, comparing Fig. 3A (top left figure with CFTR-F508del residual activity = 13.9%) and Fig. 3B (top right figure, CFTR-[F508del-I1027T] residual activity = 23.1%), we concluded that the residual activity of the CFTR protein in the case of the described complex allele is almost 10% higher.

We also checked the publications of Amaral and Sheppard laboratories and did not find fundamental research relevant to this CA, but we are very grateful for your help in finding the information necessary for this review.

After this I am hesistant to review the article further. Please make sure every reference is correct and statements are supported by correct literature.

Response: Thank you very much for bringing this to our attention. We have carefully reviewed our references and apologize for any inaccuracies that may have been present. We have taken steps to correct these errors and ensure that our references are complete and accurate. Thank you again for your valuable feedback.

Reviewer 3 Report

Comments and Suggestions for Authors

In this interesting review, the authors discussed the current application of intestinal organoids in study of the alleles irregularity from different patients, and served as an ex vivo model for the intervention study. This is a very important review that focus on a specific topic, and interesting to show the difference of intestinal organoids when alleles are mutated. 

Overall this review is good, while section 2 might be a bit misleading with the subtitle since the whole section barely discussed about the allelels dysregulation in different Intestinal organoids system. Considering the discussed maintext, use a title about the composition and applications of Intestinal organoids system in biomedical research might be more apporiate. however, the authors can also think of direct the maintext in section #2 to match the subtitle. 

Although some of the data is ok in review article. a schematic can help to the audience understand more of the topic. Eg, how is the organoids differ from each other when coming from patients with different complex alleles. 

ALso, if there any possibility to provide a small discussion with a table or figures for: how organiod system can help to study the role of different complex alleles in regulating the expression of CFTR.

Author Response

In this interesting review, the authors discussed the current application of intestinal organoids in study of the alleles irregularity from different patients, and served as an ex vivo model for the intervention study. This is a very important review that focus on a specific topic, and interesting to show the difference of intestinal organoids when alleles are mutated.

Overall this review is good, while section 2 might be a bit misleading with the subtitle since the whole section barely discussed about the allelels dysregulation in different Intestinal organoids system. Considering the discussed maintext, use a title about the composition and applications of Intestinal organoids system in biomedical research might be more apporiate. however, the authors can also think of direct the maintext in section #2 to match the subtitle.

Response: Thank you for bringing this to our attention. We have carefully reviewed the sectionand realized that the title did not accurately reflect the full picture described. We have corrected it by making the title more general.

Although some of the data is ok in review article. a schematic can help to the audience understand more of the topic. Eg, how is the organoids differ from each other when coming from patients with different complex alleles.

Response: Dear reviewer! If you mean the phenotypic difference between organoids from each other, phenotypic characteristics are present in Figure 3. Significant phenotypic differences are observed in only the case of organoids obtained from a CF people and healthy volunteer. All cultures of organoids from patients with CF are similar to each other - with a reduced lumen and irregular shape like at Figure 3.

Also, if there any possibility to provide a small discussion with a table or figures for: how organiod system can help to study the role of different complex alleles in regulating the expression of CFTR. Please edit.

Response: Dear reviewer, we have added a graphic abstract to this review. Thank you very much for taking the time to provide us with your valuable comments.

Round 2

Reviewer 2 Report

Comments and Suggestions for Authors

Authors have done a nice job revising the manuscript. I believe this is a good contribution. One minor comment:

Is there RNA-level evidence to suggest L4667F-F508del would beling to class I of CFTR classification. If this complex allele results in a significant reduction in RNA such that there is no protein made then I would accept this in class I. Otherewise state that protein is misfolded in a manner such that it is no longer a target for the correctors. There are many missense variants that do not respond to CFTR modulators like I507del.

I wish good luck to the authors for their continued interest in CF. 

Author Response

Authors have done a nice job revising the manuscript. I believe this is a good contribution. One minor comment:

Is there RNA-level evidence to suggest L467F-F508del would beling to class I of CFTR classification. If this complex allele results in a significant reduction in RNA such that there is no protein made then I would accept this in class I. Otherewise state that protein is misfolded in a manner such that it is no longer a target for the correctors. There are many missense variants that do not respond to CFTR modulators like I507del.

I wish good luck to the authors for their continued interest in CF. 

Response - We are very grateful to the Reviewer for the helpful comments! We did not find evidence that the level of RNA decreases in case with L467F-F508del, however, in the articles by E. Sondo and N. Baatallah we found data that L467F and L467F-F508del dramatically affect protein maturation and in our review we provide information from these articles in lines 238-248 and 351-361. Thanks to your comment, we have made changes to Table 2 and the text (line 341), replacing class I with class I-II.